# Enhanced Conformational Sampling of Nanobody CDR H3 Loop by Generalized Replica-Exchange with Solute Tempering

**DOI:** 10.3390/life11121428

**Published:** 2021-12-18

**Authors:** Ren Higashida, Yasuhiro Matsunaga

**Affiliations:** Graduate School of Science and Engineering, Saitama University, Saitama 338-8570, Japan; r.higashida.901@ms.saitama-u.ac.jp

**Keywords:** nanobody, complementarity determining region, CDR H3 loop, molecular dynamics simulation, replica exchange molecular dynamics, rare event sampling

## Abstract

The variable domains of heavy-chain antibodies, known as nanobodies, are potential substitutes for IgG antibodies. They have similar affinities to antigens as antibodies, but are more heat resistant. Their small size allows us to exploit computational approaches for structural modeling or design. Here, we investigate the applicability of an enhanced sampling method, a generalized replica-exchange with solute tempering (gREST) for sampling CDR-H3 loop structures of nanobodies. In the conventional replica-exchange methods, temperatures of only a whole system or scaling parameters of a solute molecule are selected for temperature or parameter exchange. In gREST, we can flexibly select a part of a solute molecule and a part of the potential energy terms as a parameter exchange region. We selected the CDR-H3 loop and investigated which potential energy term should be selected for the efficient sampling of the loop structures. We found that the gREST with dihedral terms can explore a global conformational space, but the relaxation to the global equilibrium is slow. On the other hand, gREST with all the potential energy terms can sample the equilibrium distribution, but the structural exploration is slower than with dihedral terms. The lessons learned from this study can be applied to future studies of loop modeling.

## 1. Introduction

Heavy-chain antibodies, identified in camelids and sharks, are composed of only heavy chains, unlike conventional antibodies (IgGs), which have both light and heavy chains [1]. The variable domains of a heavy-chain antibody are known as nanobodies. The nanobody is a small domain composed of approximately 125 amino acid residues, but shows more heat resistance and similar affinities to antigens compared with conventional antibodies [2]. Furthermore, a nanobody is able to reversibly refold and bind to antigens after heat denaturation [2]. Some nanobodies can bind to the antigen’s binding sites (epitopes) where conventional antibodies cannot bind. For example, Schoof et al. have recently developed nanobodies that bind the spike protein of SARS-CoV-2 in a fully inactive conformation [3]. Due to these properties, the nanobody attracts the attention of researchers as an important alternative to conventional antibodies in the fields of industrial and therapeutic applications.

In the structure of nanobody, the third complementarity determining region (CDR-H3) is the most crucial region compared with the other CDR regions (first and second), since it often acts as a binding site (paratope) for the antigen [4]. Therefore, predicting the structure of the CDR-H3 loop from the sequence in silico is an important first step for the rational design of nanobody to neutralize the target antigen. However, the diversity of lengths and sequences in CDR-H3 makes it challenging to establish the relationship between the loop structure and sequence in this region. Particularly in the case of nanobodies, the lengths of CDR-H3 are longer than those of antibodies [5], which leads to a more complicated relationship.

Recent machine learning and AI technologies have succeeded in predicting various new fold structures from co-evolutionary information and structural database information with high accuracy [6,7]. However, since the sequence-structure relationship in CDRs is not based on evolution, it is difficult to predict the structures of CDR regions using existing methodologies. For example, when using Alpha Fold 2 [6] to predict the structure of a nanobody, the confidence level (pLDDT) of the CDR-H3 loop structure outputs low values, indicating that the predicted structure in this region is not reliable (data not shown).

Molecular dynamics (MD) simulation, a semi-empirical method based on the first principle, can be applied even in this situation. MD simulations have been used for many antibodies and nanobodies and have successfully predicted structures and binding poses in several cases [8]. The difficulty in applying MD to the sampling of CDR-H3 loop structures is the slow relaxation time scales of the loop structures. In general, the formation of a loop structure is related to the transitions of backbone dihedral angles and the packing of the side chains, which results in various metastable states in the conformational space. For this reason, it is difficult to reach the slowest relaxation time to find the most stable structure with conventional/brute-force MD simulations. Therefore, various enhanced sampling methods have been applied in the studies of CDR-H3 loop structure sampling of antibodies and nanobodies. For example, Shirai et al. used the multicanonical method for sampling the CDR-H3 loop region of antibodies and succeeded in searching for stable conformational states by crossing a large free energy barrier [8]. Fernàndez-Quintero et al. used metadynamics to investigate the structural diversity of the CDR-H3 loop of antibodies [9]. Miao and McCammon applied an accelerated MD to simulate the docking process of nanobody to GPCR [10]. Other than the sampling of the loop structures or binding poses, MD simulations have been widely applied to investigate the thermal stability of nanobody [11], binding affinity estimations [12], and response of antibody to antigen binding [13].

The whole structure of a nanobody is characterized by the fact that the CDR-H3 loop structures are diverse, while the structures of the framework regions (other than the CDR) are always preserved among nanobodies, well predicted by homology modeling methods. Therefore, in MD simulations, it is sufficient to enhance only the sampling of the CDR-H3 loop region for structure prediction. To enhance the sampling of specific regions, the collective variable (CV) based approaches are often applied, in which we choose some coordinates involving the regions of interest as CVs. However, since the CDR-H3 loop region of nanobody tends to be a long sequence (more than 20 residues, for example), it is rather difficult to choose appropriate CVs even in this region [5].

Along with CV-based enhanced sampling methods, temperature-replica molecular dynamics (T-REMD) have been widely applied to sample protein structures [14]. The weakness of T-REMD is that the probability of temperature exchange rapidly decreases with *f*^1/2^ (where *f* is the degree of freedom of the system). Therefore, a large number of replicas is required when the simulation system is large. In order to overcome this problem, the extensions of the T-REMD, REST [15], and REST2 [16,17,18] were developed. These methods locally scale the potential energy of the solute molecules and exchange the solute’s scaling parameters (or solute “temperatures”) with other replicas. By focusing on the solute’s energy, the effective number of degrees of freedom is decreased and the probability of temperature exchange can be improved. However, REST and REST2 can only select the whole molecule as a solute, and cannot select a part of a molecule, such as the CDR-H3 loop region in a nanobody. The generalized REST (gREST) method, recently developed by Kamiya and Sugita [19], further extended REST2. The gREST allows us to use a more flexible selection of the “solute”. In gREST, we can select a part of the molecule as well as a part of the potential energy terms. Therefore, using gREST, we can select the CDR-H3 loop region as the solute region for replica exchange.

In this study, we will investigate the applicability of gREST to the CDR-H3 loop structure. Specifically, we select the CDR-H3 loop as a “solute” and investigate which of the potential energy terms is effective for the conformational sampling of the CDR-H3 loop. In this paper, we first introduce gREST in Section 2 and describe the computational setup for applying gREST to four nanobodies with different CDR-H3 loop structures. In Section 3, we show the gREST simulation results of the four nanobodies. Finally, we discuss the interpretation of the results and the applicability of gREST in Section 4.

## 2. Materials and Methods

### 2.1. Generalized Replica-Exchange with Solute Tempering (gREST)

To develop an efficient computational protocol for sampling the CDR-H3 loop structure, we investigate the applicability of gREST to nanobodies. The gREST is a natural extension of REST2 that extended the original T-REMD by the scaling of the potential energy of the solute molecule rather than the temperature on the entire system. In the following, we explain the details of gREST, following the description in the original paper [19].

First, the fundamental idea behind REST2 and its first version, REST, is to improve the exchange probability by focusing on a part of the system, a *solute*, and scale the potential energy term involving it, rather than changing the temperature of the entire system as in T-REMD. In REST2, the potential energy of a replica with replica-index *a* is defined as follows:(1)$$E_{m\left(a\right)}^{\mathrm{REST}2}=\frac{\beta_{m\left(a\right)}}{\beta_{0}}E_{uu}\left(x_{a}\right)+\sqrt{\frac{\beta_{m\left(a\right)}}{\beta_{0}}}E_{uv}\left(x_{a}\right)+E_{vv}\left(x_{a}\right),$$
where $E_{uu}$, $E_{uv}$, and $E_{vv}$ are the solute-solute, solute-solvent, and solvent-solvent interaction energies, respectively. $\beta_{0}=1/k_{B}T_{0}$ is the inverse temperature of the heat bath in the simulation. $k_{B}$ is the Boltzmann constant. $\beta_{m\left(a\right)}=1/k_{B}T_{m\left(a\right)}$ is the solute’s inverse “temperature” or scaling parameter of replica-index *a* where its temperature-index is denoted by *m*(*a*). The probability that the scaling parameters (or solute “temperatures”) $\beta_{m\left(a\right)}$ and $\beta_{n\left(b\right)}$ of replica-indices *a* and *b* are exchanged is determined by the Metropolis criterion as follows:(2)$$P\left(m\left(a\right)↔n\left(b\right)\right)=\{\begin{array}{cc} 1, & \Delta_{m\left(a\right)↔n\left(b\right)}\leq 0\\ \exp\left(-\Delta_{m\left(a\right)↔n\left(b\right)}\right), & \Delta_{m\left(a\right)↔n\left(b\right)}>0 \end{array}.$$

Here, $\Delta_{m\left(a\right)↔n\left(b\right)}$ is determined to satisfy the detailed balance between the states of the extended ensemble as follows:(3)$$\begin{array}{cc} \Delta_{m\left(a\right)↔n\left(b\right)}= & \left(\beta_{n}-\beta_{m}\right)\left(E_{uu}\left(x_{a}\right)-E_{uu}\left(x_{b}\right)\right)\\ & +\sqrt{\beta_{0}}\left(\sqrt{\beta_{n}}-\sqrt{\beta_{m}}\right)\left(E_{uv}\left(x_{a}\right)-E_{uv}\left(x_{b}\right)\right) \end{array}.$$

In the original T-REMD, the solvent-solvent interaction, $E_{vv}$, appears in $\Delta_{m\left(a\right)↔n\left(b\right)}$, thereby theoretically decreasing the exchange probability according to *f*^1/2^ (where *f* is the degree of freedom of the system). On the other hand, in Equation (3), $E_{vv}$ is cancelled out and disappears. For this reason, REST2 can realize replica exchanges with a smaller number of replicas.

In gREST, the potential energy of REST2 is further extended so that not only the whole solute molecule but also a part of the molecule or potential energy terms can be selected as a “solute”. The potential energy of the replica-index *a* in gREST is defined as follows:
(4)$$E_{m\left(a\right)}^{\mathrm{gREST}}=\frac{\beta_{m\left(a\right)}}{\beta_{0}}E_{uu}\left(x_{a}\right)+\sum_{i}{\left(\frac{\beta_{m\left(a\right)}}{\beta_{0}}\right)}^{\frac{k_{i}}{l_{i}}}E_{uv,i}\left(x_{a}\right)+E_{vv}\left(x_{a}\right).$$

Here, the solute-solvent interaction $E_{uv}$ in REST2 is decomposed as $\sum_{i}{\left(\frac{\beta_{m\left(a\right)}}{\beta_{0}}\right)}^{\frac{k_{i}}{l_{i}}}E_{uv,i}$, where $l_{i}$ is the maximum number of atoms involved in the *i*-th solute-solvent potential energy term, and $k_{i}$ is the number of atoms involved in the “solute” region and the *i*-th solute-solvent potential energy term [19]. In gREST, Equation (3) is changed as follows [19]:
(5)$$\begin{array}{cc} \Delta_{m\left(a\right)↔n\left(b\right)}= & \left(\beta_{n}-\beta_{m}\right)\left(E_{uu}\left(x_{a}\right)-E_{uu}\left(x_{b}\right)\right)\\ & +\sum_{i}\beta_{0}^{1-\frac{k_{i}}{l_{i}}}(\beta_{n}^{\frac{k_{i}}{l_{i}}}-\beta_{m}^{\frac{k_{i}}{l_{i}}})\times \left(E_{uv,i}\left(x_{a}\right)-E_{uv,i}\left(x_{b}\right)\right). \end{array}$$

Equation (5) reduces to $\Delta_{m\left(a\right)↔n\left(b\right)}$ of REST2 when the solute molecule and all potential energy terms are chosen as a “solute”. When the whole system is chosen as a “solute”, it reduces to $\Delta_{m\left(a\right)↔n\left(b\right)}$ of T-REMD.

Equation (5) shows that the exchange probability between replicas can be further improved by selecting a part of the molecular region rather than the whole molecule or by selecting a part of potential energy terms as a “solute”. In general, however, it is not obvious which potential energy terms are effective for sampling conformations, while the molecular region of interest is often obvious. Specifically, for sampling the CDR-H3 loop conformations of nanobody, it may be obvious to choose the H3 loop as a “solute”, while it is not clear which potential energy terms should be chosen as a “solute”. To address this issue, in this study, we will apply several choices of the potential energy terms and study which one is efficient for sampling CDR-H3 loop conformations.

### 2.2. Simulation Setup

In this study, we use gREST to sample the conformations of the nanobody CDR-H3 loop. The gREST allows us to select a part of the molecule or a potential energy term as a “solute”. We chose the CDR-H3 region (and a part of the CDR-H1 region) with several choices of potential energy terms as a “solute” to study which terms are efficient for sampling the CDR-H3 loop structure. As a feature of nanobody, most of the framework regions except for the CDR regions have a well-preserved structure and can be predicted accurately by homology modeling, while the CDR-H3 loop is known to have a diversity in the structures depending on the sequence. Our aim is to find the best computational protocol to predict the CDR-H3 loop structure as much as small computational resources by focusing on CDR-H3 loop regions (Figure 1). Regarding the potential energy terms, we evaluated dihedral terms, vdW terms, electrostatic terms, and all the potential energy terms.

We performed gREST simulations for four nanobodies with different structures in the CDR-H3 loop. The first nanobody (PDB ID: 4KRN [20]), called 4KRN hereafter, is known to bind and inhibit EGFR. The X-ray structure of 4KRN was observed in the unbound form, which is used as the target structure in gREST. The 4KRN has a CDR-H3 loop defined by Chothia numbering of 21 residues (THR99-TYR119), which is a very long loop not observed in antibodies. The X-ray structure of its CDR-H3 loop is “wrapped” towards the CDR-H2 region (Figure 1B). The second nanobody is the anti-cholera toxin VHH (PDB ID: 4IDL [21]), called 4IDL hereafter. In addition, 4IDL has a relatively long CDR-H3 region (15 residues in Chothia numbering, ASP110-SER114), but it has a beta-hairpin structure away from the VH-VL interface region of antigen, different from the structure of 4KRN. This feature allows some hydrophobic residues to be exposed to the surface in 4IDL. The third (PDB ID: 6DBA [22]) is R303, which binds at the c-Met interaction site on internalin B as a competitive inhibitor preventing bacterial invasion. Hereafter, we call R303 as 6DBA. The X-ray structure of 6DBA, resolved in its unbound form, has a relatively long CDR-H3 region (14 residues, HIS99-VAL112 according to Chothia numbering), which contains a short helical segment in the loop. A noncanonical disulfide bond is formed between CDR-H3 and CDR-H1 that links the long CDR-H3 loop against the framework region. The final nanobody is Nb23 (PDB ID: 7EH3 [23]), which inhibits the self-aggregation of very amyloidogenic variants of b2-microglobulin. Hereafter, we call Nb23 as 7EH3. The structures of 7EH3 were determined in the solution environment by NMR. In its PDB entry, the 10 best structures from energy minimization were deposited, showing rather large mobilities in the CDR-H3 region (17 residues, TYR101-LEU117). The orientation of the CDR-H3 loop remains by the cation-p interaction between ARG50 and TYR103 of the CDR-H3.

We performed gREST simulations with eight replicas to study which potential energy terms can be useful to reproduce the crystal structure in as short a simulation time as possible. First, the initial structures for simulations were created from the target-nanobody sequences by homology modeling using the SWISS-MODEL web server [24], respectively. The C-terminal of each nanobody was capped with N-methyl. Since the disulfide bond in the framework region as well as in the CDR-H3 loop for 6DBA were successfully detected in this homology modeling step, we explicitly incorporated those bonds in the setup. After energy minimization, 125 ps isothermal MD was performed at 300 K imposing positional restraints on the heavy atoms. Then, 125 ps isothermal and isobaric MD were conducted at 300 K and 1 atm with the same restraints. Thereafter, we removed all of the restraints and performed 312.5 ps isothermal MD with eight replicas applying the gREST potential energies (Equation (4)). As the solute or scaling region, we selected the CDR-H3 loop (and a part of CDR-H1 region) for gREST simulations. After the equilibration of replicas with the gREST potential energies, we optimized the scaling parameters $\left\{\beta_{m\left(a\right)}\right\}$ (or solute “temperatures”) by running 6.25 ns gREST simulation, in order for the exchange probabilities with the neighbors to become ≅0.3. We optimized the parameters by repeating the gREST exchange several times with tentative parameters, and updating the parameters after checking the deviation from 0.3. The largest scaling parameter $\beta_{1}$ remained at the same temperature as the MD’s thermostat (i.e., $\beta_{1}=\beta_{0}=1/k_{B}T$, and T = 300 K). After optimizing $\left\{\beta_{m\left(a\right)}\right\}$, we conducted a production run with the optimized parameters. Using Δt of 2.5 fs for numerical integration, the parameter exchange was attempted at every 1000 steps (2.5 ps). The simulation length of the production run was varied depending on the nanobody and the chosen potential energy term (25 to 700 ns). The above simulations were conducted using scaling dihedral terms, vdW terms, electrostatic terms, and all the potential energy terms. By “all the potential terms”, we mean the combination of dihedral, vdW, and electrostatic terms (ignoring bond, angle, UB, and CMAP terms except for 4KRN, where CMAP terms were added to all the potential energy terms).

The input file for the MD simulations were prepared using the CHARMM-GUI web server [25]. All of the MD and gREST simulations were performed using GENESIS [26,27]. CHARMM36 [28,29] was used as the force field and the TIP3P model was used for water molecules [30]. Electrostatic interactions were treated using the Smooth-PME [31], and covalent bonds including hydrogen atoms were constrained using SHAKE [32] and SETTLE [33]. The temperature was controlled using the stochastic velocity scaling method [34]. After gREST simulations, trajectories of replicas are sorted to obtain trajectories with equal scaling parameters or solute “temperatures”. Then, the trajectory with the largest scaling parameter or the lowest solute “temperature” $\beta_{1}$ was analyzed. MDToolbox.jl (https://github.com/matsunagalab/MDToolbox.jl, access date 1 December 2021) was used for most of the trajectory analysis except for DSSP [35]. The RMSD of simulation snapshots from the experimental structure was calculated using the Cα-atom Cartesian coordinates in the CDR-H3 loop.

## 3. Results

### 3.1. The gREST of 4KRN

Figure 2 shows the results of gREST simulations for 4KRN. In addition to gREST simulations with all the potential energy terms, electrostatic terms, vdW terms, and dihedral terms, we compared those with a conventional MD simulation. First, the RMSD of the conventional MD from the experimental structure (Figure 2A) remains around the initial RMSD = 5 Å even after 120 ns, which indicates that the relaxation time scale of the loop structure is very long, and hard to access with conventional MD simulations. On the other hand, the RMSDs of all the gREST simulations show large fluctuations, suggesting the efficiency of gREST simulations for exploring the loop-structure space. The RMSDs of the electrostatic terms (Figure 2C) at the first 0–40 ns are similar with those of the conventional MD, but from around 50 ns, the RMSD starts to largely fluctuate, temporarily reaching around RMSD = 2 Å, then starts to only fill the RMSD region 3–5 Å, staying away from the experimental structure. Although the result of vdW terms (Figure 2D) shows a drop of RMSD at around 170 ns, the average RMSD value looks larger than that with all the potential energy terms.

On the other hand, in the result of all the potential energy terms (Figure 2B), the RMSD gradually decreases with time, indicating that the experimental structure is successfully identified as a stable state in this gREST simulation. In fact, once the RMSD drops below 2 Å at around 140 ns, it remains with low RMSD values during 150–250 ns. The RMSD increases once at around 260 ns, but then returns to lower RMSD values at 300 ns.

Another interesting result was obtained with the gREST with the dihedral terms. In this case, the gREST quickly samples the vicinity of the experimental structure (around RMSD = 2 Å) in a short simulation time (~0.5 ns). However, the RMSD did not fall below 2 Å after that, and repeated transitions between two states (RMSD = 2 Å and RMSD = 4–5 Å) continued. Uni-directional relaxation to the experimental structure, as observed in the gREST with all the potential energy terms, was not observed. This two-state behavior is also clearly captured by the principal component analysis using the distance matrix between Cα atoms of CDR-H3 loop (Figure 2G). Then, what is the cause of this two-state behavior? This can be explained by comparing the homology-modeling structure with the experimental structure (Figure 2H,I). PRO111 in the CDR-H3 loop of 4KRN has the trans peptide bond in the experimental structure, but the homology-modeling structure has the cis peptide bond. The cis bond makes the loop structure protrude towards the molecular surface (Figure 2H). The two states sampled by the gREST with the dihedral terms correspond to these isomeric trans and cis states of PRO111.

### 3.2. The gREST of 4IDL

Figure 3 shows the results of the gREST simulations for 4IDL whose CDR-H3 has a beta-hairpin. As is well known, beta-hairpin takes a long time (more than microseconds [36]) to form, and the force field parameters are not well optimized compared with those of alpha-helix. Therefore, it is challenging to reproduce the structure with MD simulations. In 4IDL, we performed a conventional MD, gREST with all the potential energy terms, and gREST with the dihedral terms. In the conventional MD (Figure 3A), as observed in 4KRN (Figure 2A), the RMSD remains around the initial value (RMSD = 5 Å) again, indicating that the loop structure cannot be widely explored with conventional MD simulations. In the case of gREST with all the potential energy terms (Figure 3B), the structure gradually approaches the experimental structure. Again, this suggests the usefulness of gREST for sampling stable structures. However, the simulation length is still too short to find the beta-hairpin of 4IDL even with this enhanced sampling method.

The gREST with dihedral terms shows an interesting behavior again (Figure 3C). The gREST of the dihedral angle term can explore a larger conformational space than with all the potential energy terms. It successfully reaches RMSD ~ 2 Å at around 200 ns. The low RMSD values continue until around 300 ns, but the RMSD increases again after 300 ns, and the RMSD never returns to the low values (RMSD ~ 2 Å) even after extending the simulation time up to 700 ns. This suggests that, as in the case of 4KRN, the dihedral angle term gREST cannot correctly identify the experimental structure as the most stable state. As will be discussed below, perhaps the time scale of the dihedral angle transitions accelerated by gREST is too fast for other interactions (e.g., side-chain packings). Therefore, they are not able to relax to the equilibrium distributions in respective dihedral states.

### 3.3. The gREST of 6DBA and 7EH3

Figure 4 shows the results of the gREST simulations for 6DBA and 7EH3. This time, we only performed gREST simulations with all the potential energy terms and dihedral terms. The results of 6DBA are consistent with previous observations (4KRN and 4IDL) except for the fact that total RMSDs are relatively small due to the disulfide bond between the CDR-H3 and the framework region. In the gREST with all the potential energy terms, the structure gradually approaches the experimental structure, while the gREST with dihedral terms only fluctuates between metastable structures then starts to deviate from the experimental structure after 85 ns. Although the RMSD of all the potential energy terms becomes less than 2 Å at around 110 ns, the complete formation of the helix was not observed.

The results of 7EH3 show a little different behavior with the previous ones. In this case, not only the dihedral terms but also all the potential energy terms show large RMSD fluctuations. In fact, the RMSD of all the potential energy terms do not appear to be decreasing continuously, showing some jumps during 70 to 80 ns. This behavior may be explained by the large mobility of the CDR-H3 loop of 7EH3, as suggested by the previous study by Percipalle et al. [23]. The authors performed MD simulations of 7EH3 and observed rather large entire-structure RMSD values, around 2.5–3.0 Å, contributed mainly by the fluctuations of the CDR-H3 loop. These large fluctuations are related to the tentative formations/deformations of the cation-p interaction between the CDR-H3 and the framework region. Another explanation for the gREST results may be related to the properties of NMR structures. Since the NMR structures represent “snapshots” of the fluctuating structure in the solution condition, choosing the reference structure for RMSD is difficult, especially for mobile structures, such as the CDR-H3 loops. Here, we chose the 1st model in the 10 NMR structures in 7EH3. Changing the reference model did not qualitatively change the result.

## 4. Discussion and Conclusions

In this study, to develop an efficient computational protocol for sampling the CDR-H3 loop structure of a nanobody, we have performed gREST with various potential energy terms. In general, loop structures are dominated by dihedral angles, thus it is crucial to include dihedral angles in modeling [37]. For this reason, gREST with vdW or electrostatic terms was not effective for sampling the loop structure. On the other hand, gREST with all the potential energy terms always decreased the RMSD values, indicating that the gREST with all the potential energy terms correctly identifies the experimental structure as a stable state. The gREST with dihedral terms widely sampled the loop-structure space, including the experimental structure. However, the results showed large fluctuations even after reaching the experimental structure, and sometimes, it passed through the experimental structure. This suggests that the experimental structure is not identified as the most stable state in this simulation.

Why does gREST with all the potential energy terms correctly identifies the stabilities of states, while gREST with dihedral terms incorrectly identifies the stabilities? To address this question, we analyzed the gREST data across all of the replicas. Figure 5 shows the time series of scaling parameters (solute “temperatures”) for all of the replicas and the scatter plots of the RMSD and the total potential energies $E_{m\left(a\right)}^{\mathrm{gREST}}$ (Equation (4)). Figure 5A,C shows that, in 4KRN, the parameter exchanges occur more frequently in gREST with dihedral terms than with all the potential energy terms, even after optimization of the acceptance ratio at around 0.3. Moreover, the RMSD distribution of the dihedral terms shows that the clusters are apart on the RMSD space, indicating that structural changes due to the exchanges are rather large. Importantly, Figure 5B shows that for gREST with all the potential energy terms, there is a good correlation between the RMSD and the total potential energy, i.e., the lower the energy, the closer to the experimental structure. On the other hand, Figure 5D shows that there is no correlation between the RMSD and the total potential energy in gREST with dihedral terms. A similar result can be observed in the case of 4IDL (Figure 5E–H). The reason for this is that, as indicated by Figure 5C,D, in gREST with dihedral terms, the exchanges occur frequently, and the structural changes caused by the exchanges are rather large, which prevents the other degrees of freedom from relaxing to the equilibrium distribution. In fact, in the case of gREST with dihedral terms, the solute-solvent interactions have become zero, i.e., $E_{uv}$ = 0. This makes the dynamics of the dihedral angles “forget” to “feel” the interactions with the other degrees of freedom. Furthermore, de Sancho et al. showed that dihedral angles generally tend to move insensitively to the surrounding environment (such as friction [38]), which may cause memory effects resulting in slow relaxations to the equilibrium.

The lessons learned from this study will be applied to future simulation studies of the CDR-H3 loop structure. The gREST with dihedral terms can explore the global conformational space, but the relaxation to the equilibrium is slow. On the other hand, gREST with all the potential energy terms can sample the equilibrium distribution, but the structural exploration is slower than with dihedral terms. Therefore, for example, a good strategy may be to use gREST with dihedral terms *first* for global search, and *then,* perform gREST with all the potential energy terms. Using these advanced combinations in future studies, the modeling of the CDR-H3 loop structure by MD is expected to be improved.

Another important future direction is the simulation study on the structural *ensemble* of the CDR H3 loop in the solution as well as the structural changes induced by the interaction with antigens, which was not pursued in this study. The structure of the CDR H3 loop created by homology modeling often differs more than the extent of the fluctuations in the solution. In addition, this study has focused on correcting these large structural errors and predicting a single “ground-state” structure with gREST. However, since the CDR-H3 loop structures have inherently large fluctuations in the solution, knowing the structural ensemble is important for predicting paratopes. Measuring only the deviation from the X-ray structure as a reference structure is insufficient to evaluate the reproducibility of the structural ensemble of CDR-H3 loop structures, including various metastable states. This issue was well observed for 7EH3, the solution structural ensemble determined by NMR. Since 7EH3 contains the structural ensemble, comparing MD structures with RMSD from each NMR structure was not sufficient to validate the gREST applicability. In future studies, more rigorous evaluations of the reproducibility of the structural ensemble would be required, for example, by comparing the statistics of simulation trajectories with raw NMR signals. If the structural ensemble can be successfully reproduced by simulation, it would be possible to predict epitopes more accurately by applying the ensemble docking method [39]. On the other hand, if the CDR-H3 loop binds to the epitope by changing its structure through the interaction with the antigen, it is difficult to predict its binding pose by the ensemble docking method. In this case, it is necessary to perform rather extensive MD simulations including both the nanobody and antigen, as was done by Miao and McCammon [10]. In this case, the gREST MD simulations using the CDR-H3 loop and the candidate epitope regions as a “solute” would be expected to accelerate structural sampling for a better binding pose search.

## Figures and Tables

**Figure 1 life-11-01428-f001:**
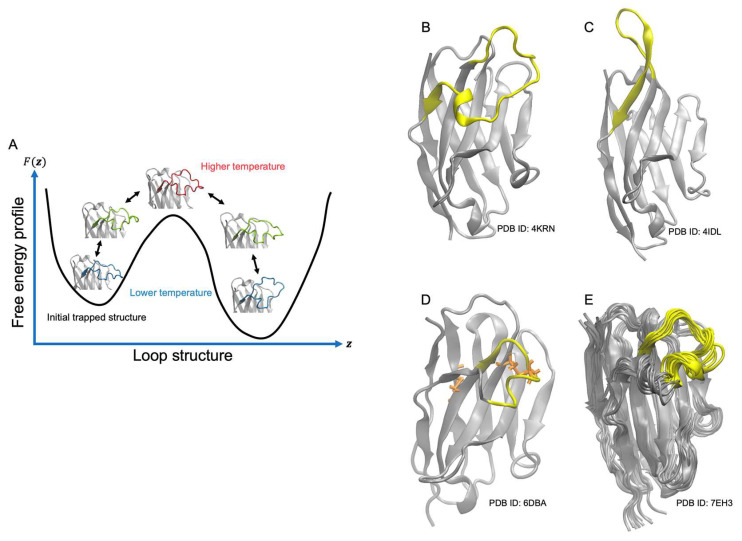
Sampling method and simulation systems in this work. (**A**) Schematic picture of gREST simulations exchanging the scaling parameters (or solute “temperature”) of the solute region (CDR-H3 loop) with other replicas. (**B**) Experimental structure of a nanobody (PDB ID: 4KRN) simulation in this work. The CDR-H3 loop is colored yellow. (**C**) Second simulation system (PDB ID: 4IDL). (**D**) Third simulation system (PDB ID: 6DBA). (**E**) Fourth simulation system (PDB ID: 7EH3).

**Figure 2 life-11-01428-f002:**
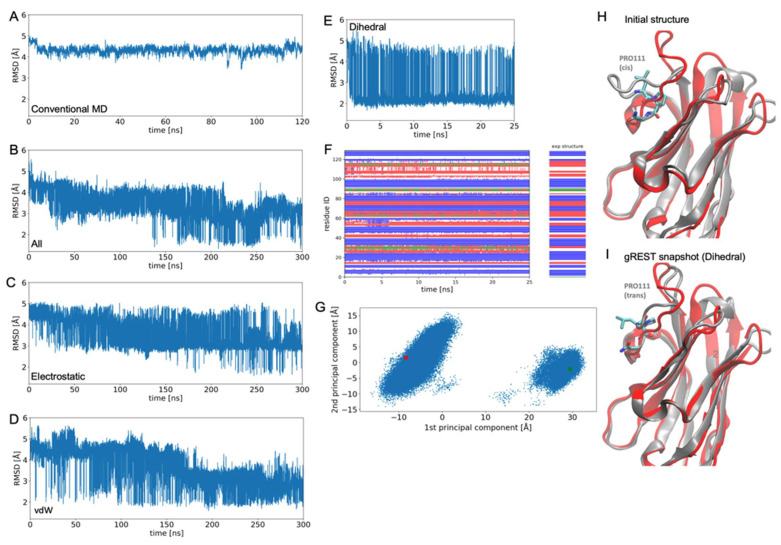
The gREST simulation trajectories of nanobody 4KRN with the largest scaling parameter or the lowest solute “temperature” $\beta_{1}$. (**A**) Time-series of RMSD from the experimental structure in conventional MD simulations. (**B**) The gREST selecting all the potential energy terms involving the CDR-H3 loop as a “solute”. (**C**) Electrostatic energy terms selected as the “solute”. (**D**) The vdW energy terms selected as the “solute”. (**E**) Dihedral energy terms selected as a “solute”. (**F**) DSSP classifications (blue: Strand, red: Loop, green: Helix) of the trajectory of gREST with dihedral energy terms. (**G**) Principal components calculated from the distance maps of Cα atoms in the CDR-H3 loop whose coordinates are obtained by gREST with dihedral energy terms. Red and green circles are the experimental and the homology-modeling structures, respectively. (**H**) The homology-modeling structure is used as the initial structure for MD and gREST simulations (grey), compared with the experimental structure (red). (**I**) Snapshot of gREST with dihedral energy terms (grey) compared with the experimental structure (red).

**Figure 3 life-11-01428-f003:**
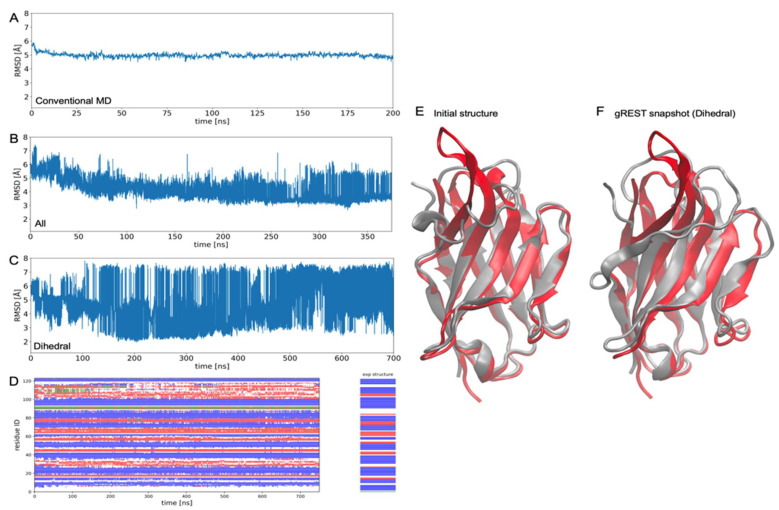
The gREST simulation trajectories of nanobody 4IDL with the largest scaling parameter or the lowest solute “temperature” $\beta_{1}$. (**A**) Time-series of RMSD from the experimental structure in conventional MD simulations. (**B**) The gREST selecting all the potential energy terms involving the CDR-H3 loop as a “solute”. (**C**) Dihedral energy terms selected as a “solute”. (**D**) DSSP classifications (blue: Strand, red: Loop, green: Helix) of the trajectory of gREST with dihedral energy terms. (**E**) Homology-modeling structure used as the initial structure for MD and gREST simulations (grey), compared with the experimental structure (red). (**F**) Snapshot (at around 200 ns) of gREST with dihedral energy terms (grey) compared with the experimental structure (red).

**Figure 4 life-11-01428-f004:**
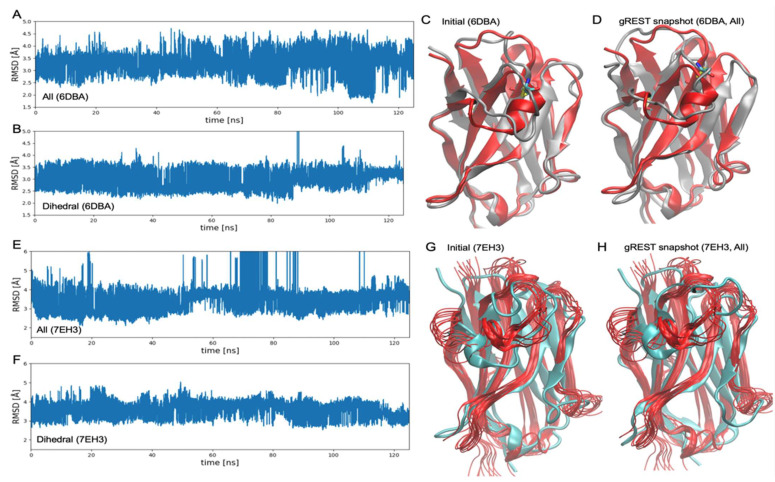
The gREST simulation trajectories of nanobodies 6DBA and 7EH3 with the largest scaling parameter or the lowest solute “temperature” $\beta_{1}$. (**A**) The gREST selecting all the potential energy terms involving the CDR-H3 loop as a “solute” for 6DBA. (**B**) Dihedral energy terms selected as a “solute” for 6DBA. (**C**) Homology-modeling structure used as the initial structure for MD and gREST simulations (grey), compared with the experimental structure (red) of 6DBA. (**D**) Snapshot (at around 120 ns) of gREST with all the potential energy terms (grey) compared with the experimental structure (red) for 6DBA. (**E**,**F**) The gREST simulation results for 7EH3. RMSDs were evaluated using the 1st model in the 10 NMR structures. (**G**) Initial structure compared with the NMR structures of 7EH3. (**H**) Snapshot (at round 100 ns) compared with the NMR structures for 7EH3.

**Figure 5 life-11-01428-f005:**
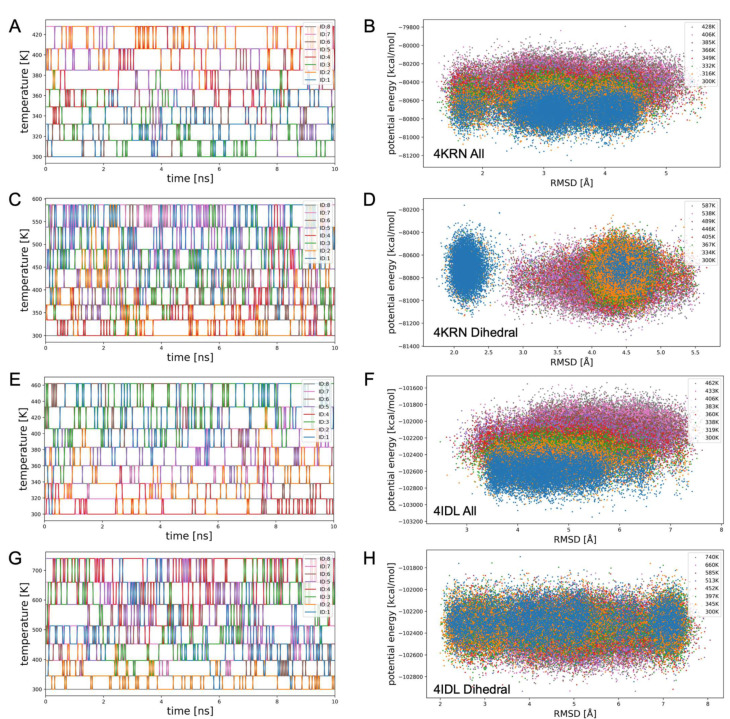
Analysis of gREST simulation data across replicas. (**A**) Time-series of scaling parameters $\left\{\beta_{m}=1/k_{B}T_{m}\right\}$ or solute “temperatures” of replicas for 4KRN. Replica-indices are indicated by different colors. (**B**) Scatter plot of the RMSD from the experimental structure and the total potential energies $E_{m}^{\mathrm{gREST}}$ (Equation (4)) of gREST. Obtained with gREST simulations of 4KRN using all the potential energy terms involving the CDR-H3 loop as a “solute”. (**C**,**D**) The gREST of 4KRN using dihedral energy terms involving the CDR-H3 loop as a “solute”. (**E**,**F**) The gREST of 4IDL using all the potential energy terms. (**G**,**H**) The gREST of 4IDL using dihedral energy terms.

## Data Availability

Input files for gREST simulations in this study are publicly available at https://github.com/matsunagalab/paper_higashida2021, access date 1 December 2021.

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
