# Peer review of "Enhanced Conformational Sampling of Nanobody CDR H3 Loop by Generalized Replica-Exchange with Solute Tempering"

_life, 2021, doi:10.3390/life11121428_

Round 1

Reviewer 1 Report

In short, the simulations seem to have been performed competently and they support the presented conclusions; actually, it is really hard for me to find any weakness of the presented reasoning. The only suggestion I have is to change the last section of the main body of the document into "Discussion and conclusions". The overall quality of the manuscript is above the average: a concise abstract, clear and readable figures, extensive work undertaken together with the relevant results obtained, good English - no typos found, etc. In my opinion the manuscript in the present form already meets requirements justifying publication and it provides significant food for thought for the readers of Life. Thus, my recommendation is to publish as it is.

Author Response

In short, the simulations seem to have been performed competently and they support the presented conclusions; actually, it is really hard for me to find any weakness of the presented reasoning. The only suggestion I have is to change the last section of the main body of the document into "Discussion and conclusions". The overall quality of the manuscript is above the average: a concise abstract, clear and readable figures, extensive work undertaken together with the relevant results obtained, good English - no typos found, etc. In my opinion the manuscript in the present form already meets requirements justifying publication and it provides significant food for thought for the readers of Life. Thus, my recommendation is to publish as it is.  

We sincerely thank the reviewer for taking the time to review our manuscript.

Following the reviewer’s comment, I have changed our section title from “Discussion” to “Discussion and conclusions”.   

Reviewer 2 Report

The idea Higashida and Matsunaga present is interesting but I think that the testing should be extended.

It is not sufficient, in my opinion, to limit the analysis to the case of only two nanobodies.

The replica exchange approach applied to the CDR3 regions only, either using all the potential energy terms or only the dihedral angle term suffers from a basic limitation: it can sample a much more extended conformational landscape of the long loop only in the solvent. This may happen to be sufficient for picking up the actual conformation for an isolated nanobody, but if the CDR3 establishes an interaction with residues of the framework region, the sampling of an extended landscape should also include the involved framework region. By the same token, the conformation of CDR3 in the complex with the antigen is most often different with respect to the structure adopted in the isolated nanobody, and the same may be true for the antigen too (at least for the epitope region). This means that an extensive exploration of the structural options leading to the proper geometry of paratope and epitope in a nanobody-antigen complex would require extending the gREST approach to the epitope region too.

On top of this basic limitation, only for isolated nanobodies there is also the possibility of divergence between the crystal structure and the actual solution structure, with the affinity and binding properties being determined by the latter. Finally, an important issue the authors do not mention at all is the fact that in most camel VHH lineages, CDR3 conformation is restricted by an additional disulfide bridge, whereas in llama VHH lineages, this is not true (by the way, llamas as well are camelids, contrary to the authors’ introductive statement).

For all these reasons, I think the authors should include more examples in their analysis. For instance, they could test the isolated nanobody structures 6DBA and 7EH3 (the second one, in particular, may be very interesting as it was obtained in solution by NMR). Moreover the comprehensive analysis of Mitchell and Colwell (ref. 5 in the manuscript), can be consulted to extract other cases where the isolated nanobody structure is available.

With these extensions and with a proper discussion on the limitations of their gREST approach for antigen-bound nanobodies, and the possible clues to mark conformational differences in solution and in crystal lattice, the paper could certainly improve and become worth of publication. 

Author Response

The idea Higashida and Matsunaga present is interesting but I think that the testing should be extended.

It is not sufficient, in my opinion, to limit the analysis to the case of only two nanobodies.

The replica exchange approach applied to the CDR3 regions only, either using all the potential energy terms or only the dihedral angle term suffers from a basic limitation: it can sample a much more extended conformational landscape of the long loop only in the solvent. This may happen to be sufficient for picking up the actual conformation for an isolated nanobody, but if the CDR3 establishes an interaction with residues of the framework region, the sampling of an extended landscape should also include the involved framework region. By the same token, the conformation of CDR3 in the complex with the antigen is most often different with respect to the structure adopted in the isolated nanobody, and the same may be true for the antigen too (at least for the epitope region). This means that an extensive exploration of the structural options leading to the proper geometry of paratope and epitope in a nanobody-antigen complex would require extending the gREST approach to the epitope region too.

On top of this basic limitation, only for isolated nanobodies there is also the possibility of divergence between the crystal structure and the actual solution structure, with the affinity and binding properties being determined by the latter. Finally, an important issue the authors do not mention at all is the fact that in most camel VHH lineages, CDR3 conformation is restricted by an additional disulfide bridge, whereas in llama VHH lineages, this is not true (by the way, llamas as well are camelids, contrary to the authors' introductive statement).

For all these reasons, I think the authors should include more examples in their analysis. For instance, they could test the isolated nanobody structures 6DBA and 7EH3 (the second one, in particular, may be very interesting as it was obtained in solution by NMR). Moreover the comprehensive analysis of Mitchell and Colwell (ref. 5 in the manuscript), can be consulted to extract other cases where the isolated nanobody structure is available.

With these extensions and with a proper discussion on the limitations of their gREST approach for antigen-bound nanobodies, and the possible clues to mark conformational differences in solution and in crystal lattice, the paper could certainly improve and become worth of publication.

First, we sincerely thank the reviewer for taking the time to review our manuscript.

As for extending the test cases: We thank the reviewer for introducing us to important test systems, 6DBA and 7EH3. In this revision, we have performed extensive simulations for these two cases and discussed the results. The results of 6DBA are consistent with the previous version of the manuscript, i.e., the gREST with all the potential energy works, but the gREST with dihedral terms does not work. On the other hand, the results of 7EH3 show different behavior from the previous ones: Both gRESTs did not show a continuous decrease in RMSD (although the value of RMSD is already relatively small ~ 3-4 Å). In the revised manuscript, we discussed this might be explained by the mobility of the CDR-H3 loop in the solution condition.

As for the interaction with antigen: We thank the reviewer for pointing out another important issue. We agree with the reviewer that interactions with antigens can affect the conformations of the CDR-H3 loops. But we would like to note that conformational samplings of isolated nanobodies, as was done in this study, provide helpful information for developing nanobody libraries. As the reviewer knows, “good” nanobody libraries should contain versatile CDR-H3 conformations. Our gREST simulations may be useful for checking conformations in developing nanobody libraries. As for the interactions with antigens, we are currently performing database work to see whether the interaction with antigen significantly changes the CDR-H3 loop structure or not. From this database work, we are selecting simulation targets and planning to perform gREST simulations of nanobody complex with antigen. Because these studies are beyond the scope of the current work, we would like to publish these ongoing studies elsewhere in the future. 

Reviewer 3 Report

The authors have presented an interesting approach to model the loop structure applying generalized replica-exchange (gREST) with solute tempering. They have examined the loop dynamics under conventional versus gREST molecular dynamics. Based on their analysis, there is practically no transition between the extended and closed-form based on the conventional method. Whereas in the proposed method, authors have found many transitions considering several reaction coordinates. Loop formation usually takes several nanoseconds (end-to-end contact formation measured around 4-6 ns). It is interesting to see the gREST method producing significant improvement.

It would be of great importance for authors to address the following questions for clarity:

  1. How do these transitions time compare to free loop versus when it is part of the whole protein? Are they slower or faster? If they are slower, then how do global dynamics contribute to the loop formation. What additional adjustments are triggering the formation pathways?
  2. It was not clear whether this analysis was based upon a single or multiple trajectories analysis?
  3. If the analysis is done from multiple trajectories, how do they vary from trajectory to trajectory?
  4. Is there an end-to-end distance distribution of the concerned loop as a function of time? If so, are they similar or different from RMSD distribution?

Author Response

The authors have presented an interesting approach to model the loop structure applying generalized replica-exchange (gREST) with solute tempering. They have examined the loop dynamics under conventional versus gREST molecular dynamics. Based on their analysis, there is practically no transition between the extended and closed-form based on the conventional method. Whereas in the proposed method, authors have found many transitions considering several reaction coordinates. Loop formation usually takes several nanoseconds (end-to-end contact formation measured around 4-6 ns). It is interesting to see the gREST method producing significant improvement.

First, we sincerely thank the reviewer for taking the time to review our manuscript.

> How do these transitions time compare to free loop versus when it is part of the whole protein? Are they slower or faster? If they are slower, then how do global dynamics contribute to the loop formation. What additional adjustments are triggering the formation pathways?

We thank the reviewer for pointing this important issue. Generally, the time scales of loop formations are very slow compared to the simulation time (as shown by conventional MDs in the manuscript). For example, the formation of the beta-hairpin of a free (isolated) peptide takes time longer than 10 microseconds, as far as we know. Also, the isomerization of proline generally occurs at slower than one millisecond. This time scale seems not to be significantly altered by the interactions with the other part of the nanobody (i.e., framework region). We successfully observed the isomerization of proline for 4KRN within nanoseconds with dihedral angle gREST simulations. Kamiya and Sugita performed 500-1000 nanosecond simulations for observing the beta-hairpin formation of a free (isolated) peptide even with gREST. Our 4IDL system (CDR-H3 contains a beta-hairpin) required comparative simulation time (700 nanosecond) for observing a partial formation of the beta-hairpin. From these observations, we think that the interactions with the framework region do not significantly alter the time scales.

> It was not clear whether this analysis was based upon a single or multiple trajectories analysis? If the analysis is done from multiple trajectories, how do they vary from trajectory to trajectory?

As for 4KRN and 4IDL, we performed two independent simulations with different initial velocities. The results were qualitatively similar, suggesting that the results are not dependent on the initial condition.

> Is there an end-to-end distance distribution of the concerned loop as a function of time? If so, are they similar or different from RMSD distribution?

We may misunderstand the reviewer’s point, but the terminal residues of the CDR-H3 loop are bonded to the framework region in nanobody. The distance between them shows only small fluctuations since the framework region of nanobody is relatively rigid. We apologize if we misunderstood the point.

Round 2

Reviewer 2 Report

I appreciate the modifications of the manuscript that were introduced by Higashida and Matsunaga who added two new nanobody models.

The most intriguing aspect of the results obtained with the newly added species appears with 7EH3 that represents the structural family of an isolated nanobody.

The application of gREST with all potential energy term or with only the dihedral angle term does not seem to reach the same result as observed with the other nanobodies. The authors first attribute this divergence to the mobility of 7EH3 CDR3 and then to the fact that the NMR structure is an ensemble and therefore it is quite hard to calculate RMSD values in the absence of a single experimental structure (therefore on the first structure was considered). The mobility of the CDR3 loop in the isolated species is not a prerogative of 7EH3. The real difference between 7EH3 and the other considered nanobodies relies in the fact that only 7EH3 is a solution structure. The other considered nanobody structures are in fact crystal structures of antigen-bound or free species. Since the conformation of the paratope is likely to change upon binding, it is surprising that the outlined method shows the same pattern of RMSD trend, with all potential energy term and only the dihedral angle term, for antigen-bound and free nanobodies. On the other hand, for 7EH3, no comparison should be possible between the final structures obtained by gREST and the experimental solution ensemble. If the method works in approaching the arrangement of the CDR3 loop in the crystal structure of the complex or of the isolated nanobody – this is something the author should explain – one should stick to the final result of gREST with all potential energy term and with a sufficient coverage of the conformational span anticipated by gREST with the dihedral angle term only and consider the extent of change the solution structure undergoes upon binding and/or crystallization.

The authors should add some comments, including also the issues above mentioned, on the results obtained with 6DBA and 7EH3 in the Discussion and conclusions section. In the revised version, this section is essentially the same as that of the original version, which is rather odd after adding the calculations of isolated nanobodies in solid and liquid state. With this addition, in my opinion the paper can be published.

Minor errors are also present: line 79, “choosing” instead of “choose”; the adjective “grey” is always misspelled as “gray”.

Author Response

> The most intriguing aspect of the results obtained with the newly added species appears with 7EH3 that represents the structural family of an isolated nanobody.

> The application of gREST with all potential energy term or with only the dihedral angle term does not seem to reach the same result as observed with the other nanobodies. The authors first attribute this divergence to the mobility of 7EH3 CDR3 and then to the fact that the NMR structure is an ensemble and therefore it is quite hard to calculate RMSD values in the absence of a single experimental structure (therefore on the first structure was considered). The mobility of the CDR3 loop in the isolated species is not a prerogative of 7EH3. The real difference between 7EH3 and the other considered nanobodies relies in the fact that only 7EH3 is a solution structure. The other considered nanobody structures are in fact crystal structures of antigen-bound or free species. Since the conformation of the paratope is likely to change upon binding, it is surprising that the outlined method shows the same pattern of RMSD trend, with all potential energy term and only the dihedral angle term, for antigen-bound and free nanobodies. On the other hand, for 7EH3, no comparison should be possible between the final structures obtained by gREST and the experimental solution ensemble. If the method works in approaching the arrangement of the CDR3 loop in the crystal structure of the complex or of the isolated nanobody – this is something the author should explain – one should stick to the final result of gREST with all potential energy term and with a sufficient coverage of the conformational span anticipated by gREST with the dihedral angle term only and consider the extent of change the solution structure undergoes upon binding and/or crystallization.

> The authors should add some comments, including also the issues above mentioned, on the results obtained with 6DBA and 7EH3 in the Discussion and conclusions section. In the revised version, this section is essentially the same as that of the original version, which is rather odd after adding the calculations of isolated nanobodies in solid and liquid state. With this addition, in my opinion the paper can be published.

We again thank the reviewer for taking the time to review our manuscript.

We agree with the reviewer that the current study is limited to the single “ground-state” structure of isolated nanobody, and we should discuss the solution structural ensemble and structural changes induced by the interaction with antigens. Following the reviewer’s comment, we added discussions on the limitation of the current study and possible future directions.

Page 11 lines 385-408, we added the following discussion:

“Another important future direction is the simulation studies on the structural ensemble of the CDR H3 loop in solution and the structural changes induced by the interaction with antigens, which was not pursued in this study. The structure of the CDR H3 loop created by homology modeling often differs more than the extent of the fluctuations in solution, and this study has focused on correcting such large structural errors and predicting a single “ground-state” structure with gREST. However, since the CDR-H3 loop structures have inherently large fluctuations in solution, knowing the structural ensemble is important for predicting paratopes. Measuring only the deviation from the X-ray structure as a reference structure is insufficient to evaluate the reproducibility of the structural ensemble of CDR-H3 loop structures including various metastable states. This issue was well observed for 7EH3, the solution structural ensemble determined by NMR. Since 7EH3 contains the structural ensemble, comparing MD structures with RMSD from each NMR structure was not sufficient to validate the gREST applicability. In future studies, more rigorous evaluations of the reproducibility of structural ensemble would be required, for example, by comparing statistics of simulation trajectories with that of raw NMR signals. If the structural ensemble can be successfully reproduced by simulation, it would be possible to predict epitopes more accurately by applying the ensemble docking method [39]. On the other hand, if the CDR-H3 loop binds to the epitope by changing its structure through the interaction with the antigen, it is difficult to predict its binding pose by the ensemble docking method. In this case, it is necessary to perform rather extensive MD simulations including both nanobody and antigen, as was done by Miao and McCammon [10]. In such a case, gREST MD simulations using the CDR-H3 loop and the candidate epitope regions as a “solute” would be expected to accelerate structural sampling for better binding pose search.”

  1. Miao, Y.; McCammon, J.A. Mechanism of the G-Protein Mimetic Nanobody Binding to a Muscarinic G-Protein-Coupled Receptor. PNAS 2018, 115, 3036–3041, doi:10.1073/pnas.1800756115.
  2. Yamashita, T. Toward Rational Antibody Design: Recent Advancements in Molecular Dynamics Simulations. International Immunology 2018, 30, 133–140, doi:10.1093/intimm/dxx077.

> Minor errors are also present: line 79, "choosing" instead of "choose"; the adjective "grey" is always misspelled as "gray".

Thank you for pointing them out. We corrected them in this revision.

Reviewer 3 Report

Simulation forcefield for hairpin is far from being optimized, and for helix, on the other hand, can fairly reliably produce observed relaxation time. For the hairpin formation time scale, authors should consult Munoz and Eaton et al. Nature article 1997. For loop formation, time scale authors may consult Eaton et al. for end-to-end contact formation applying the triplet quenching method. End-to-end contact formation measuring speed limit in protein folding by Zewail. A.H. et al. in PNAS 2011 may be examined.  

Author Response

> Simulation forcefield for hairpin is far from being optimized, and for helix, on the other hand, can fairly reliably produce observed relaxation time. For the hairpin formation time scale, authors should consult Munoz and Eaton et al. Nature article 1997. For loop formation, time scale authors may consult Eaton et al. for end-to-end contact formation applying the triplet quenching method. End-to-end contact formation measuring speed limit in protein folding by Zewail. A.H. et al. in PNAS 2011 may be examined.   

We again thank the reviewer for taking the time to review our manuscript.

Thank you for introducing us important papers. We agree with the reviewer that the force field parameters for beta-hairpin are far from being optimized compared to those for alpha-helix. In this revision, we added descriptions for the time-scale for beta-hairpin formation and the force field issue in Result of 4IDL, citing Munoz and Eaton Nature article 1997.